# 3D-Printed Dip Slides Miniaturize Bacterial Identification and Antibiotic Susceptibility Tests Allowing Direct Mastitis Sample Analysis

**DOI:** 10.3390/mi13060941

**Published:** 2022-06-14

**Authors:** Tai The Diep, Samuel Bizley, Alexander Daniel Edwards

**Affiliations:** Department of Pharmacy, Reading School of Pharmacy, University of Reading, Reading RG6 6UR, Berkshire, UK; s.c.bizley@reading.ac.uk

**Keywords:** microfluidic microbiology, antimicrobial resistance, dip slide, minimum inhibitory concentration, 3D printing, miniaturized microbiology

## Abstract

The early detection of antimicrobial resistance remains an essential step in the selection and optimization of antibiotic treatments. Phenotypic antibiotic susceptibility testing including the measurement of minimum inhibitory concentration (MIC) remains critical for surveillance and diagnostic testing. Limitations to current testing methods include bulky labware and laborious methods. Furthermore, the requirement of a single strain of bacteria to be isolated from samples prior to antibiotic susceptibility testing delays results. The mixture of bacteria present in a sample may also have an altered resistance profile to the individual strains, and so measuring the susceptibility of the mixtures of organisms found in some samples may be desirable. To enable simultaneous MIC and bacterial species detection in a simple and rapid miniaturized format, a 3D-printed frame was designed for a multi-sample millifluidic dip-slide device that combines panels of identification culture media with a range of antibiotics (Ampicillin, Amoxicillin, Amikacin, Ceftazidime, Cefotaxime, Ofloxacin, Oxytetracycline, Streptomycin, Gentamycin and Imipenem) diluted in Muëller–Hinton Agar. Our proof-of-concept evaluation confirmed that the direct detection of more than one bacterium parallel to measuring MIC in samples is possible, which is validated using reference strains *E. coli* ATCC 25922, *Klebsiella pneumoniae* ATCC 13883, *Pseudomonas aeruginosa* ATCC 10145, and *Staphylococcus aureus* ATCC 12600 and with mastitis milk samples collected from Reading University Farm. When mixtures were tested, a MIC value was obtained that reflected the most resistant organism present (i.e., highest MIC), suggesting it may be possible to estimate a minimum effective antibiotic concentration for mixtures directly from samples containing multiple pathogens. We conclude that this simple miniaturized approach to the rapid simultaneous identification and antibiotic susceptibility testing may be suitable for directly testing agricultural samples, which is achieved through shrinking conventional tests into a simple “dip-and-incubate” device that can be 3D printed anywhere.

## 1. Introduction

Rising antimicrobial resistance remains one of the biggest global threats to public health, animal health, and the environment. For example, within an agricultural setting, bovine mastitis is a major challenge to the dairy industry both economically through reduced milk production and affecting animal well-being. Antibiotic use to treat mastitis is associated with increasing antibiotic resistance. In addition, there is a great diversity of both pathogens and commensal microbial communities present in bovine mastitis [1,2]. Recent studies have indicated that microbes interacting in species and communities have contributed to individual bacterial growth and their subsequent evolution of antibiotic resistance [3,4], although the relevance of their resistance regarding selection in the microbiome is still unclear [5]. This highlights the importance of exploring the association between host–pathogen, pathogen–pathogen and pathogen–commensals, including the positive or negative impact of antibiotics on the natural microbiota [6,7]. One way to improve this understanding in the face of microbial complexity is to scale up our ability to measure antibiotic sensitivity.

Microbiology testing remains central, contributing to diagnosis and the effective treatment of infections, and surveillance to guide empirical antibiotic selection is important as well. Measuring minimum inhibitory concentration (MIC) using antibiotic susceptibility testing assays remains vital, and it is included in many standardized and clinical reference methods, with more accuracy compared to disk diffusion [8]. The MIC is still an important value for making clinical decisions. Combined with a pharmacokinetic-pharmacodynamic profile, MIC values guide the selection of the effective dose for an antibiotic [9,10]. Analysis of MIC remains critical for setting breakpoint threshold concentrations that define resistance vs. susceptibility, underpinning the surveillance of resistance levels among bacteria important to human and animal health [11]. Monitoring MIC shows new bacterial resistance trends as they emerge, and they can be used to modify clinical doses administered to achieve a required therapeutic concentration in patients or animals [12]. MIC measurements form a vital quality control and standardization tool between different laboratories across the world. Every year, MIC values are reported to reference agencies including CLSI, EUCAST, and BSAC and are used to define standards for clinical and veterinary laboratories and to assess global microbiological susceptibility trends [5,13]. 

MIC is measured by a range of antibiotic susceptibility assays in vitro, including broth microdilution, agar dilution, and antimicrobial gradient methods such as E-test (bioMerieux, Marcy–L’Etoile), MIC Test Strip (Lio-filchem Inc., Waltham, MA, USA), M.I.C. Evaluator (Oxoid, Basingstoke, UK) and Ezy MIC Strip (HiMedia Laboratories Pvt. Ltd., Mumbai, India). Many automated systems adapt these methods for higher throughput. These are classically conducted as routine methods within a microbiology laboratory, but they are associated with high costs, a need for skilled workforce with careful quality control systems and substantial labor time [14]. The interpretation of MIC results can be influenced by several factors including the concentration of bacterial inoculum, the way the concentration of antibiotics is prepared, and the quality of media. The current approach detects MIC for a single bacterial isolate and as a result typically takes 2–3 days to complete and analyze as the pure strains need to be isolated after overnight culture before performing identification and then finally antibiotic susceptibility to determine MIC. In well-resourced situations, rapid identification (e.g., MALDI-TOF) combined with rapid susceptibility testing systems can reduce this to 1–2 days. The impact of this for people without rapid identification or susceptibility testers is that it delays the physician or veterinary surgeon’s decision making, resulting in a possible delay in patient treatment. Alternatively, antibiotics are used empirically without testing. In addition, with current methods, there are a number of other factors that reduce/restrict the efficiency of the testing process, including the following: antibiotic panels of testing from commercial suppliers are not customizable, significant time is required for agar media preparation, and supply chain challenges remain regarding the import of specific antibiotics in some countries. A separate weakness of current testing protocols is a focus on single isolates that may not reflect the progress of any infection caused by a combination of multiple bacterial strains, either multiple pathogens or a single pathogen growing amongst commensal organisms that may influence antibiotic responsiveness [15]. 

To address the high level of antibiotic resistance, a variety of methods have been developed and commercialized such as clinically oriented automated systems including VITEK 2^®^ [16] and MALDI-TOF VITEK MS^®^ systems [17]. Several factors must be considered when developing suitable methods including time saving, cost saving, reliability and robustness of the method, alongside convenience for the end-user. The majority of molecular biology-based methods offer the rapid detection of specific targets including identification and a panel of common resistance genes, alongside the expansion of next-generation sequencing that has improved our understanding of microbiome and the resistome. We still rely heavily on phenotypic assays, and these continue to focus on single strains, yet the importance of understanding the response of mixed microbial populations to antibiotics is driving the development of direct sample testing methods. 

Recently, the use of digital cameras for timelapse imaging has contributed to improved analytical microbiology methods such as the early detection of antimicrobial resistance (AMR). Based on the falling cost and rapidly improving performance of digital cameras driven by consumer products (smartphones), the process of bacterial growth and response to antibiotics can be recorded without requiring large colonies clearly visible to the eye after overnight growth. Many miniaturization efforts focus on the microscopic analysis of growth exploiting digital cameras for time-resolved microscopy [18,19]. Others have directly imaged bacterial particles in liquid using digital cameras imaging fluorescence [20] or scatter [21]. Conventional culture methods have been improved through digital imaging, giving us a far more detailed understanding of colony growth on solid media [22]. These innovations have reached clinical microbiology guidelines, with EUCAST protocols allowing disc-diffusion antibiotic susceptibility to be recorded in as little as 4 h of culture [23]. 

At the same time that high-performance cheap digital cameras permit smaller colonies to be detected earlier, additive manufacturing methods such as 3D printing have become inexpensive and accessible, with fused-filament deposition methods cost-effective for producing routine microbiology labware [24]. Applying 3D printing to rapid custom microbiology labware production, we designed a new frame to create dip slides that combine two main functional goals: (1) directly identify and (2) determine the antibiotic MIC of microbes from samples. By shrinking solid culture format by >100-fold, a set of 6 mm well diameters in each dip slide frame uses far less media but still allows single colonies of bacterial species to grow and be identified. The detection of smaller colonies on these miniaturized versions of Petri dishes is aided by digital imaging using inexpensive cameras (e.g., smartphone, compact consumer camera), thereby avoiding expensive laboratory scanning instruments. The 3D-printed frame holding multiple antibiotic concentrations in agar is easy to prepare and has a fully customizable panel, making it easier to use compared to the current agar dilution methods. The aims of this study are to firstly optimize the 3D-printed frame for making dip slides for the detection of reference bacterial strains and to analyze the antibiotic susceptibility profile, to secondly observe the kinetics of bacterial growth on the different agar surfaces, and finally to evaluate the analysis of bacteria in bovine mastitis samples as a proof-of-concept application. 

## 2. Materials and Methods

### 2.1. Experimental Approach

The design of the 3D printed frame used in this study includes two parts to identify bacterial species with two 35 mm × 7 mm rectangles; and to determine MIC, we used two rows of 6 mm diameter round wells. All designed files were published as open source models for customization or 3D printing by anyone [24]. The two rows of 10× round wells include up to 10 antibiotic concentrations, which are typically prepared as doubling dilutions. The size of the round-shaped well was >150 times smaller than current conventional Petri dishes (Figure 1), allowing each sample to be tested on a large panel of conditions.

### 2.2. Bacterial Strains, Mastitis Samples, Media, and Measurement of MIC

Bovine mastitis samples were collected from the Centre for Dairy Research (CEDAR) at the University of Reading. All samples were transported to the laboratory and processed/analyzed on the collection day. Milk samples were diluted at 3 serial dilutions until 1:10^−3^, and 100 μL of these dilutions was applied to streak out onto the CHROMagarTM Mastitis agar plates prepared following manufacturers guidance (CHROMagar™, Paris, France) to identify the bacteria. Antibiotic susceptibility tests were performed by agar dilution on Muëller–Hinton Agar (MHA) (Sigma). All reference strains and isolates, including *K. pneumoniae* ATCC 13883, *E. coli* ATCC 25922, *P. aeruginosa* ATCC 10145, *S. aureus* ATCC 12600, and isolates from mastitis samples such as *Staphylococcus aureus*, *E. coli*, *Pseudomonas* spp., *Klebsiella* spp., *Streptococcus uberis*, and *Streptococcus agalactiae* were inoculated overnight to reach at 10^8^ CFU/mL in Muëller–Hinton Broth (MHB) (Sigma for microbiology, Dorset, UK). Conventional MIC measurement was performed with CLSI guideline (CLSI M100-ED31:2021) and also by the E-test methods. Examples of different specific culture media including MacConkey agar (MC), Baird Parker Agar (BP), Muëller Hinton Agar (MHA) (Sigma Aldrich, Dorset, UK), and CHROMagar^TM^ Mastitis (CHROMagar™, Paris, France) were used to identify bacteria. 

### 2.3. Bacterial Identification and MIC Measurement Using 3D-Printed Dip Slides

#### 2.3.1. Preparation of 3D Printed Dip-Slide Frame

The Standard Tessellation Language (STL) design of the frame dip slide is available online [24], the frame dips slide were 3D printed, sterilized with 70% alcohol and dried before use. CHROMagar™ Mastitis (CHROMagar™, Paris, France) was prepared according to the manufacturer’s instructions and added to long rectangles to identify bacterial species. Muëller–Hinton (MH) Agar with triphenyl tetrazolium chloride (TTC) dye added to make colonies appear more strongly in digital photographs was added with serial dilutions of antibiotics to determine MIC. The antibiotic powders of Ampicillin, Amoxicillin, Amikacin, Ceftazidime, Cefotaxime, Ofloxacin, Oxytetracycline, Streptomycin, Gentamycin and Imipenem were purchased from Sigma Aldrich (Dorset, UK). The range of 9 antibiotic concentrations prepared in MH agar for testing was from 0.125 to 32 μg/mL for each antibiotic in a doubling dilution series.

#### 2.3.2. Detection of Bacterial Species and Determining MIC Using 3D-Printed Dip Slides

After collecting, milk samples were diluted into two serial 10-fold dilutions and considered as testing samples. The dip slides were directly dipped into this diluted milk, kept submerged in the samples for 1 min, and removed, allowing excess sample to drain off the dip slide. Finally, the dip slide was placed in a sealed box to prevent drying out and incubated at 37 °C for 12 h (Figure 2). The interpreted results were based on the bacterial growth on two types of agar media. Firstly, the colony appearance on two types of commercial chromogenic and selective agars was noted for species identification—Gram-negative and Gram-positive, with different colored colonies identified following the manufacturer’s guidance. Alongside identification, a panel of antibiotic concentrations in agar was used for agar dilution determination of MIC (Figure 2). MIC results were recorded as the lowest concentration of antibiotic that inhibits the growth of bacteria. Note that the exact volume of sample deposited following dipping was not known; therefore, colonies were not counted to determine bacterial cell density. Thus, although the number of colonies could be counted visually by eye or through image analysis of digital photographs, the smaller area of solid media means that the range of concentrations that can be easily counted is smaller than for a conventional Petri dish. 

Muëller–Hinton Agar was used for the determination of MIC according to standard agar dilution methodology, and CHROMagarTM Mastitis (CHROMagar™, Paris, France) was used for identifying species. The colonies with specific colors helped to identify specific bacterial species, and at the point with and without bacterial colonies, growth was used to determine MIC breakpoint (Figure 2).

### 2.4. Time-Lapse Imaging to Check Kinetics of Bacterial Growth on Millifluidic Solid Microbiological Media 

With the same preparation as that of the dip slide, MH, MacConkey agar, and CHROMagarTM Mastitis (CHROMagar™, Paris, France) were applied to observe the kinetics of bacterial growth for *E. coli* ATCC 25922. The Raspberry camera 2.0 and Python code was used to take the photo, and ImageJ was used to build the time-lapse of bacterial growth. The distance from camera to 3D frame was 10.5 cm, and one cover with 3.5 cm height was used to prevent the agar from drying during time-lapse imaging. In other experiments, a digital Canon EOS 1300D with a Canon EF-S 60 mm f/2.8 Macro USM Lens was used to capture images representing the bacterial morphology on agar on the dip slides.

### 2.5. Comparison of MIC Breakpoints with E-Test Trip and Broth Microdilution in a Single Strain and a Mixture of Bacterial Strains

We compared our frame dip slide to broth microdilution methods. To do so, the density of inoculum was standardized at 10^8^ CFU/mL equivalence; i.e., McFarland 0.5 was used for testing with the frame dip slide and broth microdilution method. The panel of antibiotic and dye was used the same for frame and broth microdilution methods; the final volume for each test was 200 μL in 96-well plates. Both the frame dip slide and microwell were kept at 37 °C overnight for endpoint analysis (12–16 h). The MIC was read at the points where antibiotics stopped the growth of bacteria. The whole protocol to evaluate the novel dip-slide test was mapped in Figure 3. 

## 3. Results

### 3.1. Use of 3D-Printed Multiplex Dip Slide for Bacterial Identification and Antibiotic Susceptibility Testing

The 3D-printed frame allowed us to combine two types of solid media on a single dip slide. Longer wells provided enough space to identify colonies of bacterial species on specific chromogenic agar for Gram negatives or Gram positives simultaneously. Alongside this on the same device, agar dilution antibiotic susceptibility testing was possible. Even with only 6 mm diameter wells, single separate colonies were visible on these tiny solid media wells, with a similar appearance to a conventional Petri dish culture (Figure 1). Customization of the CAD design allows the user to decide the type of agar media required for testing to match the test purpose in the laboratory. Colonies visible on the miniaturized 6 mm wells could be counted by eye or from digital images, but this device was not designed for colony counting. Whilst we found testing very repeatable, the dip-slide format does not deposit a precise known volume onto the agar. Here, we focused on mastitis milk sample testing and adapted frames for two different identification agars provided in the CHROMagar™ Mastitis product (CHROMagar™, Paris, France). Using this frame, <1 mL of each type of agar is needed instead of 20 mL of agar for a standard 100 mm Petri dish. The MIC agar dilution wells require just 140 μL of media per slide, which is >70 times less than standard test tube agar dilution protocols and >150 times less than Petri dish, alongside a major reduction in incubator space. 

### 3.2. Characteristics of Multiplex Multifluidic Dip Slides

By dipping directly frame into the sample (e.g., diluted milk) and incubation at 37 °C (Figure 2), the results were able to be reported in just 8 h for *E. coli* and around 10–12 h for other bacteria such as *Staphylococcus* spp. and *Klebsiella* spp. (Figure 4). Bacterial colonies grown on the dip slide appeared comparable to those grown on a Petri dish for a wide range of agar types. However, it was significantly easier to record smaller bacterial colonies with a digital camera when using media that contained a supplement of TTC dye rather than Muëller–Hinton agar without dye, where the yellow-gray colony color was only weakly visible on the background of the agar color itself (Figure 4A).

To monitor speed of colony appearance, a Raspberry Pi equipped with the v2 camera module was used to capture images every 10 min. This time-lapse analysis showed that the results of MIC can be interpreted around 7–8 h for *E. coli* (Figure 4B) and more than 10 h for other bacteria such as *Staphylococcus aureus*, *Pseudomonas aeruginosa*. This means that the results of MIC can be interpreted during 7–8 h compared to routine methods needed overnight (Figure 4B). 

### 3.3. Detection of Individual Bacteria Isolates and Their Antibiotic Resistance Profile

Three-dimensional (3D)-printed dip slides with sets of miniaturized 6 mm diameter dip-slide wells plus rectangular identification strips were compared to conventional broth microdilution for the capacity to detect individual strains and measure their antibiotic susceptibility. Antibiotics such as Ampicillin, Amoxicillin, Amikacin, Ceftazidime, Cefotaxime, Ofloxacin, Oxytetracycline, Streptomycin, Gentamycin and Imipenem were used for this purpose, with Figure 5 showing images with four representative antibiotics. There was an agreement of MIC results between our dip slide to broth dilution methods. 

Next, bacterial isolation was compared between the agar dish and miniaturized dip slides with specific agar media—Mackonkey agar (MC) and CHROMagarTM Mastitis (CHROMagar™, Paris, France). Notably, the data indicated 100% accordance of individual strain using different agar media for testing the bacterial growth. The results showed *E. coli*—pink colonies and *Klebsiella* spp.—blue colonies grown on CHROMagarTM Mastitis (CHROMagar™, Paris, France) were the same results on the frame and agar dish (Figure 5).

As expected, although the overall pattern of antibiotic sensitivity was the same, there were some minor disagreements in MIC results between the dip slide and conventional broth dilution for some strain/antibiotic combinations. For example, Gentamycin differed by one doubling dilution of MIC (Figure 5). However, this is common for MIC determination and can result from different interpretations of intermediate growth, where it can be hard to decide which dilution constitutes inhibition vs. growth. For example, when capturing the photo, some small colonies can be seen on agar by eye, but these are not obvious on the digital photo, whereas the microplate shows intense coloration from microbial growth at this concentration, possibly resulting in the difference seen in the MIC of one doubling dilution. Despite the small surface agar (6 mm), bacterial colonies have still been separately seen, allowing the inhibition by antibiotics to be seen. This is important as after performing MIC, many resistant strains will be kept for further testing (e.g., for surveillance or reporting) research in the future such as genotype and prediction of resistant trends of bacterial collection. With such purpose, an individual colony was collected and kept in the biobank. There was no difference in picking individual colonies between tube test, Petri dish or dip-slide frame (Figure 5 and Figure 6). For example, when comparing the MIC breakpoint and the *E. coli* growth on specific agar—MC and CHROMagarTM Mastitis (CHROMagar™, Paris, France), the data showed that the *E. coli* grew on MC and CHROMagarTM Mastitis (CHROMagar™, Paris, France) with typical pink colony, and its MIC to Oxytetracycline was 1 μg/mL. At this concentration, there was no bacterial growth on this concentration. However, for Streptomycin and Amoxicillin, the MIC breakpoint was over 32 μg/mL because *E. coli* kept growing at this concentration (Figure 6). The same results have been found on the broth dilution methods with the same strains and antibiotics. With the current MIC methods, strains needed to be isolated and identified, and the purity needs to be checked before performing MIC. Alongside that, an extra test tube—just bacteria without antibiotic—was needed to conduct parallel with MIC. The main purpose for doing this is to make sure the right clone of bacteria was relevant to its MIC. Our frame, however, does not need to perform this step because resistant strains still grew on the agar media, and it was easy to collect the right clone for storage in terms of further research. 

### 3.4. Direct Testing of Mixtures of Bacteria and Their Antibiotic Resistance Profile 

In reality, bacteria live in their niches or community, not existing as pure individual strains, and this can affect both response to antibiotics and the measurement of antibiotic MIC. Many samples contain mixtures of microbes; it is rare to find a sample with only one single species or strain, although clinical samples often contain one dominant pathogen that can be readily identified, which forms the basis for established conventional microbiology testing. A second reason for testing mixtures is to reduce the time-to-result by directly testing samples without initial overnight culture and colony isolation. Therefore, it remains important to explore if antibiotic susceptibility of mixtures of pathogens could be measured. To do this, we tested mixtures of pathogens isolated from mastitis milk as well as reference strains, using a multiplex dip slide compared with broth microdilution as well as testing mixtures of reference strains as control groups. With our dip-slide frame, there were always two parts: identification and MIC breakpoints. On the identification panel, the aim was to identify different bacteria present as single colonies present on specific types of identification agar. However, the MIC was not identical if a single strain was compared to two mixed strains. To prove this concept, we performed MIC for each single strain (Table 1) and then mixed them at equal cell concentrations and performed MIC again for the mixture. In the experiment of mixture strains, the MIC results always showed the highest concentration of antibiotic resistance. For example, the MIC of amoxicillin of *S. aureus* isolated from milk samples was 16 μg/mL and was 32 μg/mL for *Pseudomonas aeruginosa* isolated from milk as well (Figure 7). When combing two strains into a mixture, the results showed an MIC of 32 μg/mL for both strains together (Table 2). These findings implicated that the final MIC value was the highest concentration of pathogens in the mixture. However, to conclude resistance or susceptibility for individual species or isolates, we need to follow guidelines such as CLSI or EUCAST that specify single colonies must be picked prior to testing.

Interestingly, when observing the interaction of bacterial mixtures to antibiotic resistance, the results showed that there was no significant difference between using the frame dip slide and broth dilution, even performing on reference strains or isolates from bovine mastitis samples. We also tested the frame with *Streptoccocus* species and found an agreement of bacterial growth on two methods. The small compact design makes this ideal for use for obligate anaerobes within an anaerobic cabinet and incubator. Furthermore, the dip slide can be applied for both single strain and mixture strains (Table 1 and Table 2). Table 1 summarizes the MIC results for single reference and pure isolated strains; the abbreviation of “MRF” for Mastitis Reading Farm was used to name individual isolates from milk samples from cows suffering from mastitis. In Table 2, the MIC results observed for dip slides vs. conventional methods are presented for various mixtures of different microbial strains and for directly tested mastitis milk samples. 

## 4. Discussion

We characterized a simple method to make and use a 3D-printed frame design that combines multiple miniaturized agar wells into a convenient dip slide. We asked if pathogens can be identified alongside MIC determination in one direct “dip-and-test” device. The proof-of-concept results presented here show it is feasible for bacterial species to be identified on miniaturized wells of chromogenic media alongside agar dilution for antibiotic susceptibility testing, with good agreement to conventional Petri dish methods. We also show that these dip slides can simultaneously measure mixed samples containing up to four different bacteria representing more complex clinical situations where multiple organisms are present in a single sample. The overall observed MIC was similar to the inhibitory concentration for the strain within the mixture with the highest inhibitory concentration. We also demonstrate the feasibility of using these “millifluidic” multiplex dip slides for the direct testing of milk samples from cows with mastitis. From such results, we suggest that the physician or the farmer can use these rapid direct test results to select an appropriate antibiotic, reducing the risk of treatment failure if the animal is infected with at least one resistant pathogen and avoiding the need for empirical use of broad-spectrum antibiotics without testing.

There is a trade-off in sampling simplicity achieved by miniaturization vs. absolute quantitation. A defined volume of sample can be plated on a large area with Petri dish culture, allowing precise determination of colony counts over a relatively high dynamic range (from a few colonies to several hundred on a 100 mm Petri dish). However, more space and media are required, plus skilled operation. In contrast, with the miniaturized culture area on the dip slide, it was less easy to quantify exact microbial load; however, identification and MIC determination were still possible with a much simpler procedure.

Likewise, direct testing represents a trade-off of speed to result vs. precisely measuring each organism present within the sample. Current methods to detect MIC breakpoints typically take two to three days mainly because the sample requires plating overnight, following which isolates are tested to identify organism, and finally antibiotic susceptibility tests are performed. By direct testing using a 3D-printed frame dip slide, we can reduce the turnaround time to overnight or even as little as 10 h culture. A major time-saving measure was avoiding isolating individual bacterial strains before performing the antibiotic susceptibility test; instead, the ‘bulk’ resistance of directly diluted samples is estimated. Whilst for some samples containing mixtures of organisms and/or pathogens, this could lead to false resistance, it is unlikely that false susceptibility will be observed, as when mixtures were tested, the MIC was always equivalent to the organism with the highest MIC, as would be expected. This reduces the risk of false susceptibility results that could lead to more significant clinical error, in which an ineffective antibiotic is selected. By combining two routine assays—the identification and determination of MIC—into one test with the frame dip slide, it should often be possible to interpret MIC through identification, as breakpoint resistance values can vary between Gram-negative vs. Gram-positive species. Digital imaging allows recording smaller colonies at earlier timepoints than conventional overnight incubation, again if the risk of missing slower-growing organisms is taken into account.

Facing relentless increases in antimicrobial resistance, many studies have investigated innovative alternative methods and technologies for antibiotic susceptibility testing and/or resistance detection including both genotype and phenotype [25]. However, most genotype-based methods are expensive (especially sequencing) and not easy to deploy widely in all laboratories. Detection methods such as LAMP and PCR can be fast but rely on the detection of specific resistance targets that can vary over time and location. For these reasons, phenotypic assays remain crucial techniques and contribute an important central role in the determination of antibiotic susceptibility for surveillance and treatment, especially the quantitation of MIC that can be used to identify breakpoint values for resistance vs. susceptibility [26]. As a result, significant advances in phenotypic assays have been continually developed such as microplate-based surface methods [27], nanoliter array [28], and multiplex fluidic chip [29]. Our 3D-printed dip slides add to this evolution, with an intermediate level of miniaturization between microfluidics and conventional large culture dishes.

Recently, significant focus has turned to the resistome and the relationship between the complex microbiome and its interaction with pathogen behavior, and whether the overall resistance of mixtures of bacteria found in the site of infection might affect the effectiveness of antibiotic treatment [13,30]. In this study, we explore the possibility of an alternative approach to the phenotype-based method that determines the MIC value directly from mixed bacterial population in a sample [3,9,31]. Whilst this may not provide the same standardized measure of antibiotic susceptibility of each individual strain present in the sample, it will be important to continue researching the link between “mixed MIC” found from direct susceptibility testing

Our assay is simpler than many microfluidic devices, and the open source 3D-printed frame design is flexible and customizable. The system is compact, allowing mobile operation especially if combined with other portable microbiology tools such as mobile incubators [32]. Three-dimensional (3D) printing has become cheaper; we used 3D printers costing £200–£750 (Creality Ender 3 and Prusa i3 MK3) with the materials cost per frame dip slide significantly under £2 to determine simultaneously Gram-negative and Gram-positive bacterial identification plus two different antibiotics. However, in spite of being simple and inexpensive, they do need to be freshly prepared due to the limited stability of antibiotic in agar medium, and challenges remain such as scaled-up mass-production or methods to make the addition of agar to the dip slides less laborious. Future innovations should focus on exploring ways to stabilize antibiotics within such devices, permitting longer-term storage, perhaps by a combination of the dip-slide format with antibiotic discs.

## 5. Conclusions

With this simple miniaturization of essential phenotypic microbiology testing, the initial identification and MIC profiles can be simultaneously detected in direct samples. Three-dimensional (3D)-printed open source designs make this “millifluidic microbiology” accessible across the world. Further insight into the antibiotic susceptibility of mixed organisms in some important samples may also be gained by permitting a larger-scale direct phenotypic testing of complex samples. We showed that the 3D-printed dip slide was able to be used for identification and MIC determination for a range of bacteria. The dip frame is easy to use and can be easily transport to the field, with no extensive training or introduction to the user required. Combing the dip frame with our previously developed mobile incubator [32] will increase the portability of these tests for broader application, and it may reduce the turnaround time for samples taken a long way from microbiology testing labs. Finally, direct testing with 3D-printed dip slides may contribute to an alternative and rapid way to measure mixed microbes including the microbiome alongside pathogens and measure the antibiotic resistance profiles associated with complex mixtures.

## Figures and Tables

**Figure 1 micromachines-13-00941-f001:**
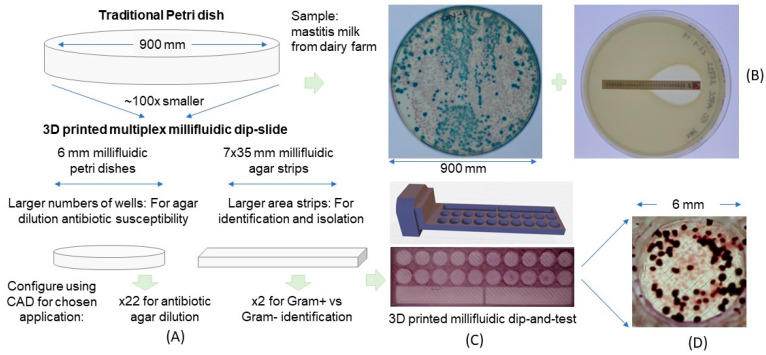
**Shrinking Petri dishes**. **Miniaturized 3D-printed arrays for solid microbiological media.** (**A**) Illustration of how a conventional Petri dish can be shrunk into a frame designed with different areas depending on required function. Smaller circular wells are used for antibiotic agar dilution, whereas longer strips are better suited to identification and colony isolation. (**B**) Examples of identification of bacteria and antibiotic susceptibility testing using a conventional Petri dish. (**C**) CAD illustration and photograph of 3D-printed frame for customized multiplex dip slide. (**D**) Zoomed-in image showing small bacterial colonies grown on individual wells of the 3D printed dip-slide frame.

**Figure 2 micromachines-13-00941-f002:**
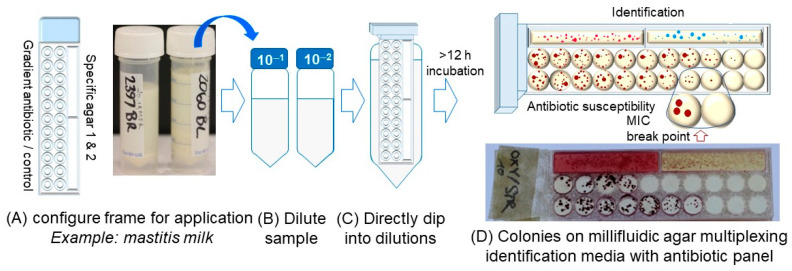
**Dip-slide testing method.** (**A**) Configure frame for application and mastitis milk samples. (**B**) Dilute sample into 2 fold, (**C**) Directly dip frame into dilutions, (**D**) Observe the results on frame.

**Figure 3 micromachines-13-00941-f003:**
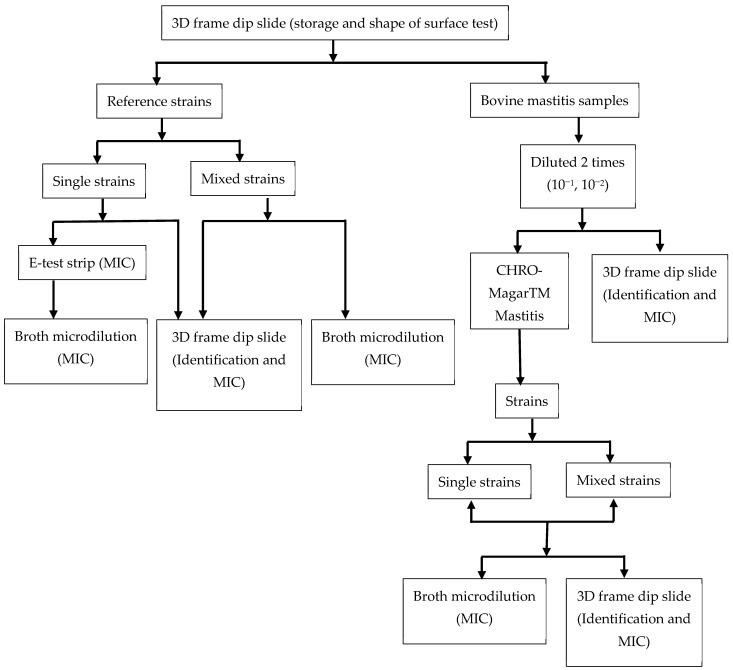
**Flowchart showing isolation, identification and sensitivity testing for reference and mastitis strains**.

**Figure 4 micromachines-13-00941-f004:**
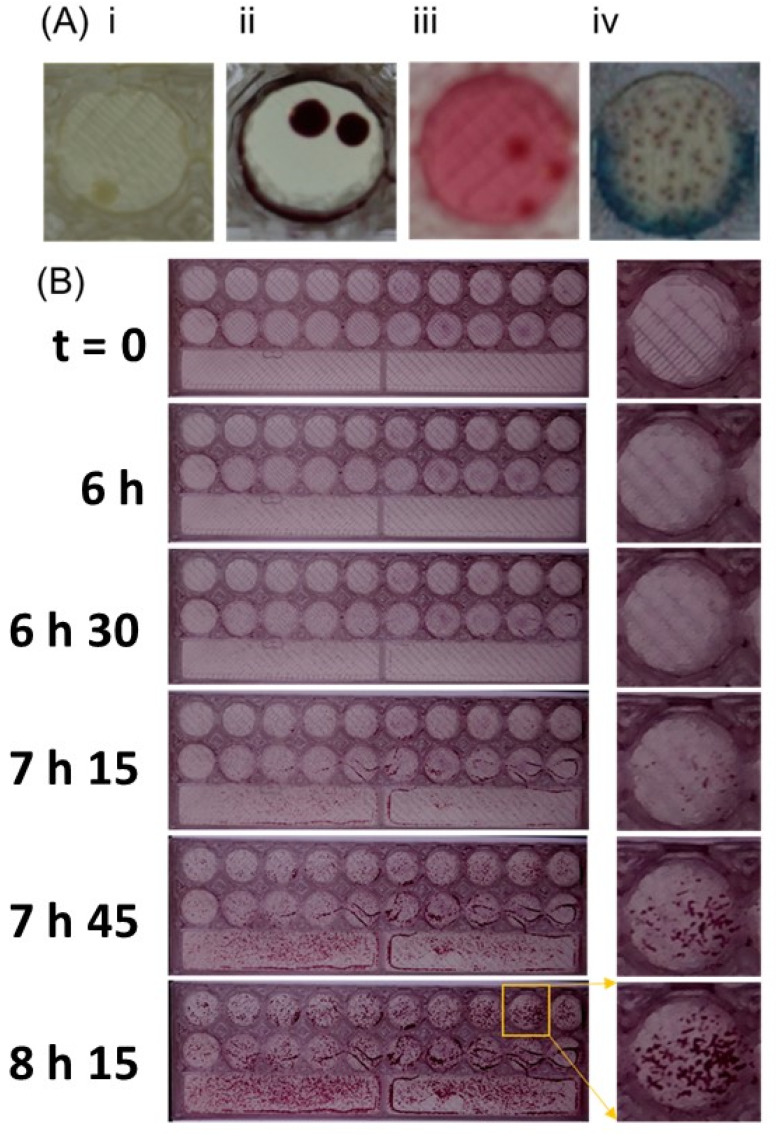
**Different media appearance and kinetics of growth on millifluidic solid microbiological media**. (**A**) The appearance of colonies on 6 mm 3D-printed discs containing the following agar types: Muëller–Hinton Agar (MHA) (i), MHA with triphenyl tetrazolium chloride (TTC) dye added (ii), MacConkey (iii), and CHROMagar™ Mastitis Gram^–^ (iv). (**B**) Selected timepoints from time-lapse analysis of colony growth, showing the emergence of colonies around 7–8 h on MHA with TTC.

**Figure 5 micromachines-13-00941-f005:**
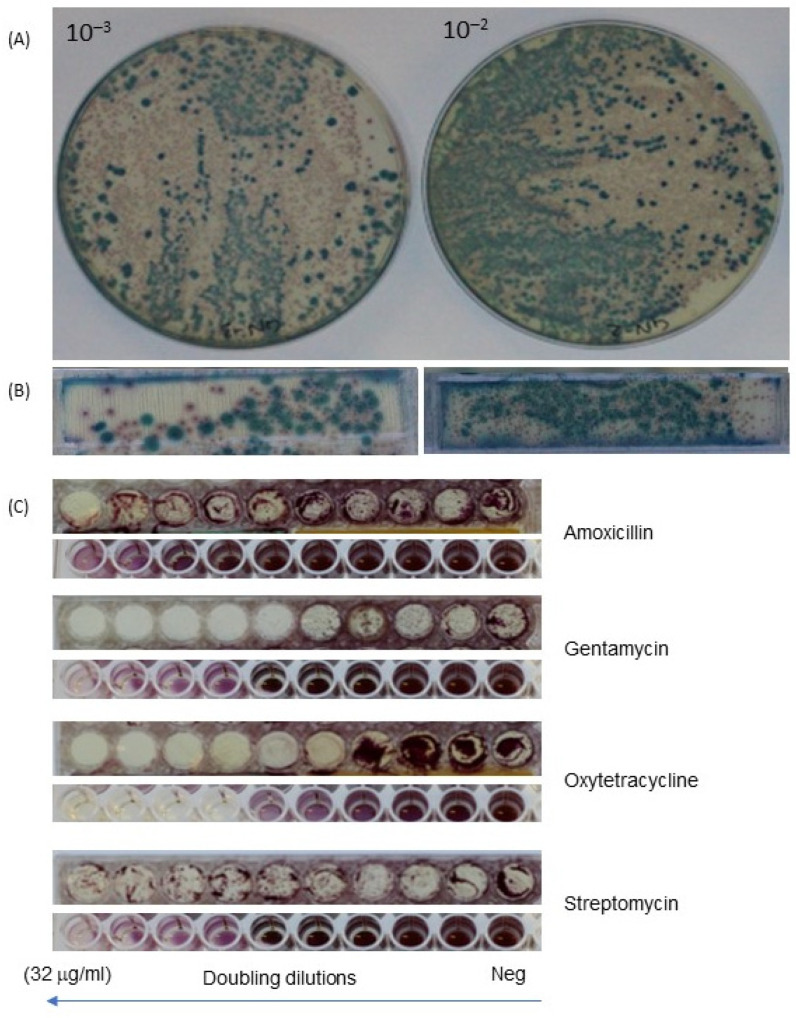
**Comparison of identification and determination of MIC through frame dip slide vs. Petri dish combined with broth microdilution on mastitis milk sample**. (**A**). Bacteria (*E. coli*—pink colonies, *Klebsiella*—blue colonies) grown on CHROMagar™ Mastitis (CHROMagar™, Paris, France) at two dilutions on Petri dish, (**B**) Bacteria grown at two dilutions on frame, (**C**) Comparison of the results of MIC between frame (upper image for each antibiotic) and broth microdilution (lower image).

**Figure 6 micromachines-13-00941-f006:**
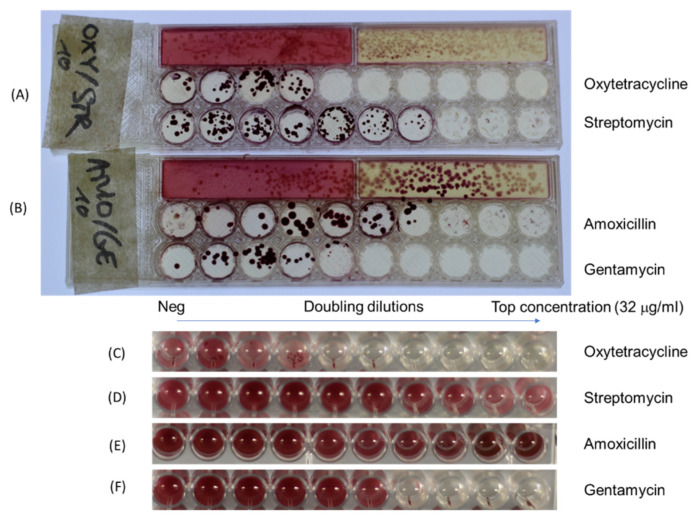
**Comparison of frame dip slide and broth dilution methods to determine *E. coli* isolated from the bovine sample and its MIC**. (**A**) The results of dip-slide tests of *E. coli* on MC/CHROMagar™ Mastitis Gram negative and its MIC for Oxytetracycline and Streptomycin, (**B**) The results of *E. coli* on MC/CHROMagar™ Mastitis Gram negative and its MIC for Amoxicillin and Gentamycin, (**C**) The results of *E. coli* on broth microdilution of Oxytetracycline, (**D**) The results of *E. coli* on broth microdilution of Streptomycin, (**E**) The results of *E. coli* on broth microdilution of Amoxicillin, (**F**) The results of *E. coli* on broth microdilution of Gentamycin.

**Figure 7 micromachines-13-00941-f007:**
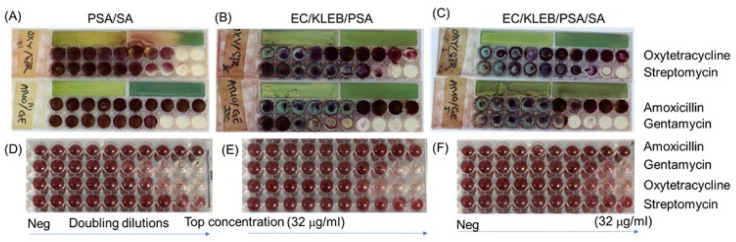
**Antibiotic susceptibility by agar dilution for mixtures of microbes, simulating multiple organisms present in some samples**. Mixtures of the following strains were prepared and split for testing in parallel using 3D-printed multiplex millifluidic dip slides (**A**–**C**) vs. conventional microplate broth microdilution (**D**–**F**). Naming of microbial species: *Pseudomonas aeruginosa* (PSA), *Staphylococcus aureus* (SA), *E. coli* (EC), *Klebsiella* spp. (KLEB).

**Table 1 micromachines-13-00941-t001:** Comparison frame dip slide vs. broth microdilution to individual strains.

Testing Samples	Methods	Antibiotic MIC (mg/mL)									
Reference Strains		AMP	AMO	AMI	CEF	CEFO	OFL	OXT	STR	GEN	IMI
*Escherichia coli* ATCC 25922 (EC)	Frame	4	>32	0.5	16	>32	2			1	0.125
	MIC broth	4	>32	0.5	16	>32	2			1	0.125
	E—strip test	4								1	0.125
*Klebsiella pneuminae* ATCC 13883 (KB)	Frame	>32	>32	1	16	>32	2			1	0.5
	MIC broth	>32	>32	1	16	>32	2			1	0.5
	E—strip test	>32								1	0.5
*Pseudomonas aeruginosa* ATCC 10145 (PSA)	Frame	>32	>32	2	16	>32	2			0.5	1
	MIC broth	>32	>32	2	16	>32	2			0.5	1
	E—strip test									0.5	1
*Staphylococcus aureus* ATCC 12600 (SA)	Frame	<0.125	>32	1	16	>32	2				
	MIC broth	<0.125	>32	1	16	>32	2				
	E—strip test	0.032									
**Direct from milk mastitis samples**											
19.02 MRF (EC)	Frame		>32					1	>32	4	
	MIC broth		>32					1	>32	4	
19.03 MRF (*Klebsiella* sp.—K)	Frame		>32					2	16	4	
	MIC broth		>32					2	16	4	
19.07 MRF (SA)	Frame		>32					>32	>32	4	
	MIC broth		>32					>32	>32	4	
19.08 MRF (SA)	Frame					16		8			
	MIC broth					16		8			
19.09 MRF (SA)	Frame		>32					16	>32	8	
	MIC broth		>32					16	>32	8	
19.09 MRF *(Klebsiella* sp.)	Frame	>32	>32	>32	16	4	2				
	MIC broth	>32	>32	>32	16	4	2				
19.10 MRF (EC)	Frame		>32					1	>32	2	
	MIC broth		>32					1	>32	2	
19.10 MRF (*Streptococcus uberis*)	Frame		>32			16					
	MIC broth		>32			16					
19.11 MRF (*Streptococcus agalactiae*)	Frame		8			16					
	MIC broth		8			16					
19.14 MRF (PSA)	Frame		>32			>32		1			
	MIC broth		>32			>32		1			
19.15 MRF (*S. aureus*—SA)	Frame		16			>32		2			
	MIC broth		16			>32		2			
19.16 MRF (EC)	Frame		16			8		2	>32	>32	
	MIC broth		16			8		2	>32	>32	
19.16 MRF (SA)	Frame		>32					1	8	16	
	MIC broth		>32					1	8	16	
19.17 MRF (*Streptococcus agalactiae*)	Frame		8			16					
	MIC broth		8			16					
19.17 MRF (SA)	Frame		>32					8	16	4	
	MIC broth		>32					8	16	4	
20. 20 MRF (*S. aureus*—SA)	Frame		8			32					
	MIC broth		8			32					
20. 20 MRF (PSA)	Frame		>32					>32	>32	8	
	MIC broth		>32					>32	>32	8	

**Table 2 micromachines-13-00941-t002:** Measure the impact of mixtures pathogens to antibiotic resistance on reference strains and mastitis strains.

Testing Samples	Methods	Antibiotic MIC (mg/mL)
AMP	AMO	AMI	CEF	CEFO	OFL	OXT	STR	GEN	IMI
**Reference Strains**											
EC/SA/KB (Reference strains)	Frame	>32	>32	2	16	>32	2				
MIC broth	>32	>32	2	16	>32	2				
EC/PSA/KB (Reference strains)	Frame	>32	>32	2	16	>32	2	>32	>32	>32	
MIC broth	>32	>32	2	16	>32	2	>32	>32	>32	
EC/SA/PSA/KB (Reference strains)	Frame	>32	>32	2	16	>32	2	>32	>32	>32	
MIC broth	>32	>32	2	16	>32	2	>32	>32	>32	
**Strains Isolated from mastitis samples**											
19.15 MRF (SA)/20.20 MRF (PSA)	Frame		>32					>32	>32	8	
MIC broth		>32					>32	>32	8	
19.17 MRF (EC)/19.09 MRF (KB)/20.20 MRF (PSA)	Frame		>32					>32	>32	>32	
MIC broth		>32					>32	>32	>32	
19.17 MRF (EC)/19.15 MRF (SA)/20.20 MRF (PSA)/19.09 MRF (K)	Frame		>32					>32	>32	>32	
MIC broth		>32					>32	>32	>32	
19.17 MRF (EC)/19.09 MRF (K)	Frame		>32					16	>32	>32	
MIC broth		>32					16	>32	>32	
19.17 MRF (*Streptococcus agalactiae*) + 20.20 MRF (*S. aureus*—SA)	Frame		8			>32					
MIC broth		8			>32					
19.10 MRF (*Streptococcus uberis*) + 19.15 MRF (*S. aureus*—SA)	Frame		>32			>32					
MIC broth		>32			>32					
**Direct from milk mastitis samples**											
19.07 MRF	Frame		>32					>32	>32	4	
MIC broth		>32					>32	>32	4	
19.09 MRF	Frame	>32	>32	>32	16	4	2	16	>32	8	
MIC broth	>32	>32	>32	16	4	2	16	>32	8

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
