# Peer review of "3D-Printed Dip Slides Miniaturize Bacterial Identification and Antibiotic Susceptibility Tests Allowing Direct Mastitis Sample Analysis"

_micromachines, 2022, doi:10.3390/mi13060941_

Round 1

Reviewer 1 Report

See attached.

Author Response

Response to review: Manuscript ID: micromachines-1745004

We thank the reviewers for their constructive suggestions and have made all requested changes to improve the clarity and accuracy of the manuscript significantly. We highlighted in yellow all significant changes. Several paragraphs and the discussion were extensively re-drafted, and so for clarity we have not included tracking of all minor edits in the tracked changes version.

Reviewer 1:

  1. Page 2 paragraph 4, the introduction of imaging based methodology seems to be incomplete, without detailed references. The use of time-lapse imaging is very popular in AST tests, and thus should be expanded in this introduction.

Added references 18-23 for this section and expanded by describing a couple of specific examples where digital imaging and time-resolved analysis have significantly improved AST.

  1. The caption of Figure 1 should be expanded, providing more information.

Expanded the figure legend to better explain the figure, making it more straight forward to view and understand (including labelling sections A, B, C, D)

  1. Page 8, why is Figure 7 before Figure 5?

Order of figures has been adjusted to make more sense.

  1. As the frame was imaged with the camera, this reviewer thinks that maybe quantifying the number of colonies from the images could be done and shown in the figures. This could help the readers to confirm MIC directly.

Response to comment: neither microbial identification nor agar dilution antibiotic susceptibility to determine MIC requires colonies to be counted. Furthermore, when dip-slides are used the volume of sample, whilst repeatable, is not precise. We have made this clear and noted that dip-slides are less suited to quantification of cell concentrations.

Further explanation and clarification of this point was added in section 2.3.2 of methods and 3.1 results.

Reviewer 2 Report

This study describes a new approach to detecting bacteria and their susceptibility to antibiotics from samples, quickly and simultaneously. A 3D printed slide dive, economical and easy to use, allows you to choose antibiotics to test and improve the clinical application.

The topic of the study is relevant and falls in the scopes of the journal. However, some sections need to be improved: in the results section the ideas do not always flow logically. The experimental data are incomplete and not well presented, they are too vague and generic: I suggest expanding the description of results in tables 1 and 2.

Discussion should be reorganized introducing a small preface at the beginning of section and explained. In addition, the concepts of sensitivity and resistance should be better used by the authors.

The English language and the clarity of the whole paper require further revision. The paper is written in a simple language, sometimes not very scientific: the text was in some points difficult to understand.

In my opinion, the manuscript is not suitable for publication in its current form: major revisions are needed.

Some specific comments

 Title

-        please add "from mastitis samples" because authors only spoke of increased antibiotic resistance in veterinary setting in introduction

Abstract

-        please explain sentence “…as the requirement of a single strain of bacteria to perform the test”:   maybe the authors would say that standard commercial test is only available for a single strain and not for a pool of bacteria?

-        please change “Muller Hinton” to “Muëller Hinton” in all paper sections

-        please add identification code of reference strains

-        please tested antibiotics name of antibiotics tested

Introduction

page 2

-        please explain what means “Furthermore, the adaptation of antibiotic resistance or the minimum selective concentration related to the MIC value, …. “

-        please rewrite in two sentences  “The current approach detects MIC for a single bacterium and takes approximately least 3 days to complete and analyze as the pure strains need to be isolated before performing MIC, the impact of this is that it elongates the physician’s/veterinary surgeons decision making time, resulting in a possible delay in patient treatment”.. It’s too long

-        please explain what means “……single isolates have little relationship to infectious disease (15)”

-        please explain AMR abbreviation

Materials and methods

page 3

-        please change to uppercase the first letter of “Petri” in all paper sections

-        please change 2.2 paragraph title from “Strains, samples, and media” to “Bacterial strains and detection of MIC”

-        please move “Bovine mastitis samples…..to identify the bacteria” to the beginning of the paragraph

-        please replace “chromoagar” with “CHROMagarTM Mastitis (CHROMagar™, Paris, France)” and add “ specific media such as MacConkey agar (MC), Baird Parker Agar (BP)and Muëller Hinton Agar (MHA) (Sigma)” before” to identify the bacteria”.

-        please delete “and their antibiotic test was followed CLSI protocol guidelines”

-        please replace “all test strains “ with “reference strains”

-        please mark all different bacteria genus and species in italics

-        please add “spp.” after pseudomonas and klebsiella genus

-        please replace “Then, the protocol of MIC was applied as CLSI guideline for all tests (CLSI M100-ED31:2021)” with “MIC evaluation was performed with CLSI guideline (CLSI M100-ED31:2021) and E-test method”

-        please add “and identification with dip slide frame” to 2.3 paragraph title

    page 4

-        please explain STL abbreviation

-        please add what types of agar were used in dip slide frame to detect Gram positive and Gram negative bacteria

-        please delate “and taking photo” from 2.4 paragraph title

    page 5

-        please change “McFarlane”  to “McFarland”

-        please replace figure 3 title with “Flowchart showing isolation, identification and sensitivity testing for reference and mastitis strains”

-        please change to uppercase the first letter of “Gram”

    page 6

-        please add “spp.” after staphylococcus and klebsiella genus

-        please delete “Bacteria that grew on the frame dip slide were as similar as on the petri dish”

  page 7

 Figure 4 caption

-        please explain TTC abbreviation

-        please change McConkey to MacConkey

-        what type of agar was used in fig 4B? Please add it.

  page 8

-        in the text the authors must first describe the results of the data in Figure 7 (Comparison of identification and determination of MIC through ..) and then those in figure 5 (Comparison of frame dip slide and broth dilution methods to determine E.coli isolated from the bovine sample and its MIC)

-        please move figure 5 before of 7: so, figure 7 becomes 5 and figure 5 becomes 6

-        please add the concentration of antibiotics to all wells in your photos (5, 6 and 7) otherwise the interpretation of the results is difficult

-        figures 7A and 7B are too dark and the colours are not clearly distinguishable

 page 9

-        add in 7C description “up” after “frame” and “down” after “microdilution”

-        please delete “petri test” and  “which single colony was also easily collected “ in the text

 page 10

-        please mark in bold the title of figure description

  page 11

-        please change the numbering of the figure from 6 to 7 and explain abbreviations (PSA/SA, EC/KLEB/PSA….) in figure description.

-        please explain sentence”the MIC of amoxicillin of S.aureus isolated from milk samples was 16 microg/ml and was 32 microg/ml for Pseudomonas aeruginosa isolated from milk as well (Figure 6)”…in fig. 6 authors tested a mix of S. aureus and P. aeruginosa and not individual strains

-        please explain sentence “When combing two strains into a mixture, the results showed a MIC of 32 microg/ml for both strains together (Table 2)”…but if is a mixture how did authors know the MIC of single strain?

-        please delete “In addition, comparison between bacterial isolation on identification agar by routine petri dish with frame dip slide were found to have the same results (Figure 7)”. It’s redundant.

-        “This frame also used to test with anaerobic bacteria such as Streptoccocus species..”.Streptococcus spp. is a facultative anaerobe bacteria so it can also grow in the presence of oxygen

-         “… the dip slide can be applied for ….as well as aerobic and anaerobic bacteria” it’s not correct (see above)

-        please explain MRF abbreviation in tables 1 and 2

Discussion

   pages 13-14

-        please add a a small preface at the beginning of discussion

-        from lines 8 to 12:  authors should pay attention to the meaning of words used!...There is confusion about scientific words. MIC was the low concentration of antibiotics that killing a mix population and not a “concentration resistance”…

-        please explain what means “…lower concentration resistance”? It is better to say that bacteria are more susceptible to that particular antibiotic at that concentration.

-        line 17: please delete figure 2 in line 17

-        lines 53-54: this dip slide was not tested on obligate anaerobe bacteria but only on a facultative bacteria (see above)

-        line 58: please delete “figure 6”

   page 15:

-        line 68: dip slide is not indicate for strict anaerobes (see above)

References

-        line 93: please mark in italic Escherichia coli

-        line 106: please delete “table of contents”

-        line 115: please mark in italic Escherichia coli and Trueperella pyogenes

Author Response

Response to review: Manuscript ID: micromachines-1745004

We thank the reviewers for their constructive suggestions and have made all requested changes to improve the clarity and accuracy of the manuscript significantly. We highlighted in yellow all significant changes. Several paragraphs and the discussion were extensively re-drafted, and so for clarity we have not included tracking of all minor edits in the tracked changes version.

Reviewer 2: 

The topic of the study is relevant and falls in the scopes of the journal. However, some sections need to be improved: in the results section the ideas do not always flow logically. The experimental data are incomplete and not well presented, they are too vague and generic: I suggest expanding the description of results in tables 1 and 2.

Discussion should be reorganized introducing a small preface at the beginning of section and explained. In addition, the concepts of sensitivity and resistance should be better used by the authors.

The discussion was restructured and rewritten to highlight the key aspects that are discussed.

The English language and the clarity of the whole paper require further revision. The paper is written in a simple language, sometimes not very scientific: the text was in some points difficult to understand. 

We have extensively edited the language use and improved clarity throughout. Furthermore we addressed all the specific comments below, to ensure particular elements in the original manuscript that were hard to understand are clearer.

Some specific comments

Title

  1. please add "from mastitis samples" because authors only spoke of increased antibiotic resistance in veterinary setting in introduction

Change title for clarity and to add specific reference to mastitis sample testing

Abstract

  1. please explain sentence “…as the requirement of a single strain of bacteria to perform the test”:   maybe the authors would say that standard commercial test is only available for a single strain and not for a pool of bacteria?

Changed wording in abstract and elsewhere to be much clearer about difference between testing isolates vs testing mixtures of bacteria.

  1. please change “Muller Hinton” to “Muëller Hinton” in all paper sections

Ctrl F Muller Hinton in the text and change to Muëller Hinton

  1. please add identification code of reference strains

Add the reference strain numbers to Abstract.

  1. please tested antibiotics name of antibiotics tested

Add list of antibiotics tested to the abstract

Introduction

page 2

  1. please explain what means“Furthermore, the adaptation of antibiotic resistance or the minimum selective concentration related to the MIC value, …. “

This paragraph was reworded extensively for clarity, and to emphasise several relevant points-  that different lab groups conduct the tests differently so MIC measurement is vital for standardisation; plus the clinical importance of MIC for dosing and surveillance.

  1. please rewrite in two sentences  “The current approach detects MIC for a single bacterium and takes approximately least 3 days to complete and analyze as the pure strains need to be isolated before performing MIC, the impact of this is that it elongates the physician’s/veterinary surgeons decision making time, resulting in a possible delay in patient treatment”.. It’s too long

? make two sentences  “

This paragraph has been extensively redrafted for clarity, to separate the time consequence of testing isolates vs the clinical significance of resistance of mixtures of organisms.

  1. please explain what means “……single isolates have little relationship to infectious disease (15)”

Again, this paragraph has been extensively reworded to improve clarity and to separate the different points.

  1. please explain AMR abbreviation

Have expanded this first use of term antimicrobial resistance (AMR)

Materials and methods 

page 3

  1. please change to uppercase the first letter of “Petri” in all paper sections

Check all text and figure and change to upper case “Petri”

  1. please change 2.2 paragraph title from “Strains, samples, and media” to “Bacterial strains and detection of MIC”

Titles of sections 2.2 and 2.3 changed to clarify and address this comment.

  1. please move “Bovine mastitis samples…..to identify the bacteria” to the beginning of the paragraph

This change was made as suggested.

  1. please replace “chromoagar” with “CHROMagarTM Mastitis (CHROMagar™, Paris, France)” and add “ specific media such as MacConkey agar (MC), Baird Parker Agar (BP)and Muëller Hinton Agar (MHA) (Sigma)” before” to identify the bacteria”.

Followed reviewer advice to change text, improving clarity.

  1. please delete “and their antibiotic test was followed CLSI protocol guidelines”

Deleted from current section but moved reference to CLSI guidelines in section 2.3.2

  1. please replace “all test strains “ with “reference strains”

Change test to ‘reference strains and isolates’ in the text.

  1. please mark all different bacteria genus and species in italics

Changes all bacteria genus and species text to italics.

  1. please add “” after pseudomonas and klebsiella genus

Add spp to the text where relevant

  1. please replace “Then, the protocol of MIC was applied as CLSI guideline for all tests (CLSI M100-ED31:2021)” with “MIC evaluation was performed with CLSI guideline (CLSI M100-ED31:2021) and E-test method”

Text changed to reviewers suggestion “MIC evaluation was performed with CLSI guideline (CLSI M100-ED31:2021) and E-test method”

  1. please add “and identification with dip slide frame” to 2.3 paragraph title

Added “using 3D printed dip slides” to the end of the 2.3. title 

     page 4

  1. please explain STL abbreviation

Define STL. Put in brackets after ……..(STL)

  1. please add what types of agar were used in dip slide frame to detect Gram positive and Gram negative bacteria

Media types used in dip-slides is now clearly stated in section 2.3.1.

  1. please delate “and taking photo” from 2.4 paragraph title

Delete ‘and taking photo’

     page 5

  1. please change “McFarlane”  to “McFarland”

Corrected spelling of this name as suggested

  1. please replace figure 3 title with “Flowchart showing isolation, identification and sensitivity testing for reference and mastitis strains”

Change figure legend to reviewers suggestion

  1. please change to uppercase the first letter of “Gram”

Change to ‘Gram’ check rest of text to make sure this is correct throughout.

Done ADE

     page 6

  1. please add “spp.” after staphylococcus and klebsiella genus

Add spp as suggested.

Done ADE

  1. please delete “Bacteria that grew on the frame dip slide were as similar as on the petri dish”

Reworded text – “Bacterial colonies grown on the dip slide appeared comparable to the those grown on a Petri dish”

Done ADE

  page 7

  Figure 4 caption

  1. please explain TTC abbreviation

TTC defined triphenyl tetrazolium chloride (TTC) in figure legend.

Done ADE

  1. please change McConkey to MacConkey

Change text to MacConkey (please check rest of text to make sure they all match.

Done ADE

  1. what type of agar was used in fig 4B? Please add it.

Add name of all agars used in Figure 4

Done ADE

   page 8

  1. in the text the authors must first describe the results of the data in Figure 7 (Comparison of identification and determination of MIC through ..) and then those in figure 5 (Comparison of frame dip slide and broth dilution methods to determine E.coliisolated from the bovine sample and its MIC)

-        please move figure 5 before of 7: so, figure 7 becomes 5 and figure 5 becomes 6

Raised by reviewer 1 also, order to be adjusted.

Done ADE

  1. please add the concentration of antibiotics to all wells in your photos (5, 6 and 7) otherwise the interpretation of the results is difficult

Add labels showing concentrations of antibiotics used to all three of these figures

Done ADE

  1. figures 7A and 7B are too dark and the colours are not clearly distinguishable

We tried to improve clarity by increase the brightness of the image (but were careful not to distort the image)

DONE?

  page 9

  1. add in 7C description “up” after “frame” and “down” after “microdilution”

Added as suggested to the figure legend “(upper image)” and “(lower image)” to be clear

DONE ADE

  1. please delete “petri test” and  “which single colony was also easily collected “ in the text

Text changed to ‘Petri dish’ rather than test, remove ‘which single colony was also easily collected’ from the text as suggested

Done ADE

  page 10

  1. please mark in bold the title of figure description

Bold figure legend title.

Done

  page 11

  1. please change the numbering of the figure from 6 to 7 and explain abbreviations (PSA/SA, EC/KLEB/PSA….) in figure description.

Check figure ordering as figure 6 comes after figure 7 similar to the 5-7 figures noted earlier. Add abbreviations as suggested to the text.

Done

  1. please explain sentence”the MIC of amoxicillin of aureusisolated from milk samples was 16 microg/ml and was 32 microg/ml for Pseudomonas aeruginosa isolated from milk as well (Figure 6)”…in fig. 6 authors tested a mix of S. aureus and P. aeruginosa and not individual strains

Reword sentence following review guidance

Done

  1. please explain sentence “When combing two strains into a mixture, the results showed a MIC of 32 microg/ml for both strains together (Table 2)”…but if is a mixture how did authors know the MIC of single strain?

Explain that the single strain was evaluated separately

Add sentence to explain “To prove this concept, we performed MIC for each single strain, then mixed them into one solution and performed MIC again”

  1. please delete “In addition, comparison between bacterial isolation on identification agar by routine petri dish with frame dip slide were found to have the same results (Figure 7)”. It’s redundant.

Remove suggested sentence from text.

Done

  1. “This frame also used to test with anaerobic bacteria such as Streptoccocus species..”.Streptococcus spp. is a facultative anaerobe bacteria so it can also grow in the presence of oxygen

Clarified as requested- deleted reference to anaerobic growth.

The same below.

  1. “… the dip slide can be applied for ….as well as aerobic and anaerobic bacteria” it’s not correct (see above)

Added explanation that it would be suitable for adaptation to anaerobic culture vessel, but removed any mention of anaerobic growth.

DONE- ADE

  1. please explain MRF abbreviation in tables 1 and 2

This abbreviation is explained now in the text in reference to mastitis samples taken from Reading Farm.

Done.

Discussion

 pages 13-14

  1. please add a a small preface at the beginning of discussion

The discussion was fully re-written into a logical set of topics, which should now be clear for the reader to follow.

  1. from lines 8 to 12:  authors should pay attention to the meaning of words used!...There is confusion about scientific words. MIC was the low concentration of antibiotics that killing a mix population and not a “concentration resistance”…

This section was significantly edited for precision and clarity.

  1. please explain what means “…lower concentration resistance”? It is better to say that bacteria are more susceptible to that particular antibiotic at that concentration.

Change to explanation reviewer has suggested.

  1. line 17: please delete figure 2 in line 17

Remove figure 2 from the text as suggested.

Done

  1.  lines 53-54: this dip slide was not tested on obligate anaerobe bacteria but only on a facultative bacteria (see above)

No further mention of anaerobic growth condition as advised, only that it would be possible to use these devices in anaerobic facilities if required.

  1.  line 58: please delete “figure 6”

Remove from text as suggested.

Done

    page 15:

  1.   line 68: dip slide is not indicate for strict anaerobes (see above)

See anaerobic points above; significant changes made to this point.

References

  1.  line 93: please mark in italic Escherichia coli

References were updated using Endnote and so reflect publisher online records.

Done

  1. line 106: please delete “table of contents”

References were updated using Endnote and so reflect publisher online records.

Done

  1. line 115: please mark in italic Escherichia coli and Trueperella pyogenes

References were updated using Endnote and so reflect publisher online records.

Done

Round 2

Reviewer 2 Report

The manuscript is now improved. I appreciate that the authors have restructured the discussion: I only suggest to delete the sentences related to the cost of the 3D printed dip slide (lines 71-74). Therefore, the manuscript requires minor changes to be published.